# Development of Saturated Fat Replacers: Conventional and Nano-Emulsions Stabilised by Lecithin and Hydroxylpropyl Methylcellulose

**DOI:** 10.3390/foods11162536

**Published:** 2022-08-22

**Authors:** Jansuda Kampa, Richard Frazier, Julia Rodriguez-Garcia

**Affiliations:** Department of Food and Nutritional Sciences, University of Reading, Whiteknights, Reading RG6 6DX, UK

**Keywords:** hydroxypropyl methylcellulose (HPMC), lecithin, butter, rheology, viscoelasticity, temperature ramp test, texture properties, stability

## Abstract

The combination of two emulsifiers, lecithin and hydroxypropyl methylcellulose (HPMC), into emulsions is an interesting strategy to design fat replacers in food matrices. The objective of this study was to investigate the effect of HPMC type and concentration on the formation, stability, and microstructure of conventional emulsions and nanoemulsions. Two different types of HPMC with low and high content of methyl and hydroxypropyl groups (HPMC-L and HPMC-H) were evaluated. The results showed that the molecular structure and concentration of HPMC play a major role in the viscoelastic behaviour, the gelation temperature, and the strength of gel formed. The firmness and work of shear of HPMC solutions increased significantly (*p* < 0.05) with increasing concentration. HPMC-L illustrated a more stable gel structure than the HPMC-H solution. Nanoemulsions showed lower moduli values, firmness, and work of shear than conventional emulsions due to the influence of high-pressure homogenization. A combination of lecithin and HPMC improved the physical and lipid oxidative stability of the emulsions, presenting a lower creaming index and thiobarbituric acid reactive substances (TBARS). In conclusion, HPMC-L at 2% *w*/*w* could be a suitable type and concentration combined with lecithin to formulate a saturated fat replacer that could mimic butter technological performance during food manufacturing operations.

## 1. Introduction

The consumption of saturated fatty acids has adverse health effects, promoting cardiovascular disease (CVD) [1,2]. One interesting strategy to decrease saturated fat in food products is through food reformulation, by replacing saturated fat with unsaturated fat; however, this is very challenging due to the specific technological functionality of saturated fat and the higher susceptibility of unsaturated fat to lipid oxidation, which can lead to an undesirable flavour profile, texture, shelf life, and loss of nutritional quality in food products [3,4].

In recent years, structured emulsions have been developed to be used as effective fat replacers in different food products by providing the desirable texture and organoleptic properties. Structured emulsions are oil-in-water systems where oil droplets are encapsulated in the layers of hydrated gelators [5]. These structures are made of a blend of liquid oils and gelators providing a semi-solid structure with higher viscosity and leading to desirable physical and organoleptic properties [5,6]. Structured emulsions such as wax-based oleogels, monoglyceride-based emulsions, and cellulose emulsions have been used as fat replacers in bakery products such as muffins and biscuits [7,8,9,10]. However, in order to improve the physicochemical stability of unsaturated oils, there has been an increasing interest in formulating nanoemulsion as lipid-based delivery systems with improved physicochemical stability of functional compounds, texture, and fatty acid profile [11,12,13,14]. Nanoemulsions with oil droplets with a particle size of less than 200 nm [13], which have a particle size smaller than conventional emulsions could provide several potential benefits in food processing such as improving physical stability to the gravitational separation of the fat phase in the matrix [15], changing the physical properties and sensory perception of the product [6,16], and improving physicochemical stability of functional compounds and unsaturated oils [13,14]. Thus, the use of nanoemulsions as delivery systems of oils is proposed in this study as a novel strategy to improve the stability of unsaturated oils as saturated fat replacers.

The combination of the oil component and the emulsifier plays an important role in the formation of the nanoemulsion. Several studies reported on the effect of oil type, with different fatty acid chain lengths, and the effect of fatty acid profile emulsion stability and bioaccessibility of lipophilic bioactive [17,18,19,20,21]. Other works have been carried out on the effect of emulsifier type and concentration on the physical and chemical stability of bioactive compounds such as β-carotene and vitamin D of nanoemulsions [17,21,22]. Moreover, combining emulsifiers into emulsions can improve the formation, stability, and functional properties of emulsions [23]. The addition of certain polysaccharides in emulsions can increase the thickness of the interface around the droplets and are able to improve the oxidative stability of the emulsion [24,25]. However, when combining emulsifiers, the formation and stability of emulsions be impacted due to the formation of hydrophobic, electrostatic, steric, or hydrogen bonding between molecules [23].

Hydroxypropyl methylcellulose (HPMC) is a hydrocolloid with surface activity, mechanical, and thermo-reversible gelation properties [26,27]. There are several studies conducted on the thermo-reversible gelation of HPMC aqueous solutions [26,28,29,30] that showed that the molecular weight with different content of methoxy and hydroxypropyl substitution and concentration of HPMC played an important role in the behaviour of solutions, the strength of gels, and the gelation temperature. Moreover, there were some studies focusing on using HPMC for improving formation, physicochemical properties, and stabilities of emulsions, which could be used as saturated fat replacers [31,32,33]. The addition of HPMC into emulsions can improve the stability of emulsions by inhibiting droplet aggregation; HPMC forms a three-dimensional network in the continuous phase limiting droplet coalescence [23,34]. Emulsion stabilised with HPMC showed greater thermal stability, showing no significant changes in the microstructure in the heating and cooling cycles [31]. Lecithin is an amphiphile molecule derived from sn-glycero-3 phosphate [4,35]. To our knowledge, there are no studies on nanoemulsion formulation considering the use of an emulsifier mixture with lecithin and HPMC to produce a stable and functional fat replacer. Therefore, in this study, we aim to understand the effect of the combination of HPMC and lecithin in the design of nanoemulsions as saturated fat replacers. The objective of this study was to examine the influence of degrees of substitution and concentration of HPMC on the rheological and texture properties of its aqueous solutions compared to butter and to investigate the effect of HPMC concentration on the formation, rheological properties, physical, and chemical stabilities of nanoemulsions and conventional emulsions formulated with lecithin and extra virgin olive oil. The results of this study will provide useful information about the basis to formulate saturated fat replacers with specific technological functionalities.

## 2. Materials and Methods

### 2.1. Materials

Two different types of commercial hydroxypropyl methylcellulose (HPMC) were used (Biosynth carbosynth Limited, Compton, UK); one with low methoxy and hydroxypropyl content (HPMC-L) and another one with higher content of methoxyl and hydroxypropyl (HPMC-H) (Table 1). Butter (salted butter with 82.0% fat and 1.5% of salt; Co-operative Group Ltd., Reading, UK) was used as reference material. Emulsions were elaborated with extra virgin olive oil (EVOO) (14.26% of saturated fat, 77.69% of monosaturated fat and 8.04% of polyunsaturated fat; Napolina brand, Co-operative Group Ltd., Reading, UK), and soy lecithin (Louis Francois Co., Marne la Vallee, France).

### 2.2. Reagents and Standards

Trichloroacetic acid, thiobarbituric acid and 1,1,3,3-tetraethoxypropane (TEP) (Sigma-Aldrich Co., Ltd., Gillingham, UK), and hydrochloric acid (Fisher scientific Co., Ltd., Loughborough, UK) were used. High purity water was used for the preparation and dilution of reagents.

### 2.3. Preparation of HPMC Solutions

Concentrations of HPMC solutions were selected after preliminary work was carried out to achieve HPMC solutions that exhibit viscoelastic moduli values within the same range. HPMC solutions (4, 6, 8, and 10% *w*/*w* of HPMC-L and 2, 3, 4, and 5% *w*/*w* of HPMC-H) were prepared following the method described by Liu, et al. [36] and Ding, et al. [37] with some modifications. HPMC powder was dispersed into distilled water using a high-speed homogeniser (Silverson, Model L4RT, Chesham, UK) at 200 rpm for 30 min at room temperature, then left at 4 °C for at least 24 h in order to allow the HPMC to be completely hydrated before the measurements. All samples were prepared in triplicate.

### 2.4. Preparation of Conventional Emulsions and Nanoemulsion Stabilised by Lecithin and HPMC

The emulsion preparation procedure was based on the methods described by Arancibia, et al. [38] and Taha, et al. [39] with some modifications. Emulsions were prepared in two steps. Firstly, a magnetic stirrer (Model SS3H, ChemLab, Barnsley, UK)) was used to prepare the aqueous phase dispersing soy lecithin (5% *w*/*w*) in water (85% *w*/*w*) at 200 rpm for 30 min at ambient temperature to ensure complete dispersion. Then, the oil was added (10% *w*/*w*) to the aqueous phase during continuous stirring. Secondly, the emulsions were homogenised with a high-speed homogeniser (Model L4RT, Silverson, Chesham, UK) at 10,000 rpm for 5 min to produce conventional emulsions (CE) or for 10 min to produce nanoemulsion (NE). The obtained CE and NE were further processed into lecithin and HPMC-L stabilised CE and NE. After assessing HPMC thermogelation properties (Section 3.1.2), it was concluded that HPMC-L presented higher mechanical strength during the cooling process, thus could have higher potential to be used as a structuring agent in complex emulsions for bakery products. Therefore, HPMC-L was selected for comparing the properties and stabilities of conventional and nanoemulsions.

HPMC-L was then added at three different concentrations (0, 2, and 4% *w*/*w*) to conventional emulsions (CE-0, CE-2, CE-4, respectively). The mixtures were stirred using a magnetic stirrer (ChemLab, Model SS3H) at 200 rpm for 3 h at ambient temperature to ensure complete dispersion, then left at 4 °C for at least 24 h in order to allow the HPMC to be completely hydrated before the measurements [36,37].

HPMC-L was added at two different levels (0% and 2%) to NE (NE-0 and NE-2, respectively). The NE was processed through a high-pressure homogeniser (HPH) (8.30H, Rannie, APV, Denmark) at 400 bars for 1 cycle [40]. This step was carried out to obtain emulsions with a mean droplet diameter lower than 200 nm [13]. It was not possible to produce a NE with 4% HPMC, as the emulsion viscosity was too high to be processed in the HPH. All samples were left for 2 h at 19 °C before measurements were performed. All emulsions were prepared in triplicate.

### 2.5. Interfacial Tension Measurement

The interfacial tension between HPMC solutions and extra virgin olive oil was determined using a pendant drop analyser (DS4270, Krüss GmbH, Hamburg, Germany) at 20 °C. An axisymmetric drop (20 µL) of HPMC dispersion was delivered and allowed to stand at the tip of the needle inside a quartz container of extra virgin olive oil (9 mL) for 15 min to achieve emulsifier adsorption at oil–water interface [41,42]. Three analytical repetitions of each measurement were conducted for each emulsion batch. Following the interfacial tension, different concentrations of HPMC solutions at 0.00, 0.001, 0.05, 0.1, 0.3, 0.5, 0.8, and 1.0% *w*/*w* were prepared to measure critical micelle concentration (CMC). CMC was determined at the intersection points of the interfacial tension values versus the HPMC concentration (logarithm) plot [43,44]. All measurements were performed in triplicate.

### 2.6. Rheological Measurements

The rheological measurements were performed using a rheometer (Anton Paar MCR 302, Anton Paar, Graz, Austria) equipped with a Peltier temperature control device. A serrated parallel plate geometry was used; the diameter of the lower stationary plate (PPTD 200/56/1) and upper plate (PP50/P2) was 50 mm; the gap between the plates was 1 mm. The samples were allowed to rest in the measurement position for 5 min for sample relaxation and temperature equilibration before measurements. Amplitude sweep and frequency sweep measurements were performed at 20 °C for HPMC solutions and emulsions, and at 30 °C for butter. In bakery products, shortening is usually incorporated into the mixing process at 25 °C, to achieve a more plastic and spreadable behaviour [45]. All the measurements were conducted in duplicates in two batches of each HPMC solution and emulsion. The storage modulus (G’) and loss modulus (G”) values were recorded.

#### 2.6.1. Amplitude Sweep

The strain sweeps were carried out at a strain amplitude range of 0.001 to 1000% and at a constant frequency of 1 Hz in order to determine the linear viscoelasticity region (LVR) of HPMC solutions and emulsions. The LVR was identified where G’ and G” were not influenced by applied strain.

#### 2.6.2. Frequency Sweep

Frequency sweeps were conducted from 0.1 to 100 Hz at a constant strain amplitude of 1.0% for HPMC solutions and emulsions, and 0.01% for butter, in order to determine G’ and G” values.

#### 2.6.3. Temperature Ramp Viscoelasticity

Temperature ramp viscoelasticity was determined according to the method of Sanz, Laguna, and Salvador [34] and Yoo and Um [26]. Temperature ramp analyses were conducted in HPMC solutions with a constant strain amplitude of 1.0% and a frequency of 1.0 Hz. Two temperature ramps were performed from 20 °C to 90 °C and 90 °C to 20 °C with a heating and cooling rate of 1 °C/min and no waiting time between the two ramps. The storage modulus (G’), the loss modulus (G”) and the complex shear modulus (G*) were recorded. The complex shear modulus was calculated as G* = G’ + i·G”.

### 2.7. Textural Analysis

A TA-XT2 texture analyser equipped with the Texture Exponent software (Stable Micro systems Ltd., Godalming, UK) was used to determine the firmness and spreadability of the butter, HPMC solutions, and emulsions. Spreadability was determined according to the method of Stefan and Kocevski [46], Chetana, et al. [47], and Glibowski, et al. [48]. Spreadability was measured using a TTC spreadability rig (HDP/SR) attachment and a 5 kg load cell. Approximately 10 g of the sample was filled into a female cone (90° angle) avoiding bubble occlusion and levelling up the surface. The male cone (90° angle) penetrated the samples 23 mm at a speed of 3 mm/s and the samples were allowed to flow outward at a 45° angle between the male and female cone surfaces. Spreadability was measured at 20 °C for HPMC solutions and emulsions and at 30 °C for butter. The parameter recorded was the work of shear (N·s) calculated from the area under the force curve. The tests were performed in triplicate for each solution and emulsion batch.

### 2.8. Measurement of Emulsion Mean Droplet Diameter (MDD), Polydispersity Index (PDI), and Particle Charge (Zeta Potential)

The particle size and polydispersity index of emulsions were determined by a dynamic light scattering (DLS) instrument (Zetasizer Nano ZS, Malvern Instruments Ltd., Worcestershire, UK) following the method of Guerra-Rosas, et al. [49] and Sharif, et al. [50]. Emulsions were diluted 100-fold with deionized water and agitated to avoid multiple light scattering effects. The dispersion was decanted into polystyrene cuvettes for measuring MDD and PDI, and placed in a disposable zeta cuvette (DTS 1070, Malvern Instruments, Worcestershire, UK) for measuring zeta potential at a wavelength of 633 nm at 25 °C. All the measurements were performed in triplicate for each emulsion batch, resulting in a total of nine replicates per sample.

### 2.9. Creaming Index (CI)

The creaming index (%) was evaluated based on the method reported by Arancibia, Navarro-Lisboa, Zúñiga, and Matiacevich [38] with some modifications. 10 mL of each emulsion were poured into a glass tube and stored at 4 °C and 20 °C. The total height (mm) of the emulsion and the cream layer were measured with a digital calliper after 1, 7, and 14 days. The Cl (%) was calculated using the following equation (Equation (1)):(1)$$\mathrm{Creaming}\;\mathrm{index}\;\%=\left(\frac{Hc}{Ht}\right)\times 100\;$$
where *Ht* is the total height of the emulsion (mm) and *Hc* is the height of cream layer (mm). Measurements were performed in triplicate.

### 2.10. Determination of Thiobarbituric Acid Reactive Substances (TBARS)

TBARS were determined according to the method of Qiu, et al. [51] and Sharif, Williams, Sharif, Khan, Majeed, Safdar, Shamoon, Shoaib, Haider, and Zhong [50] with some modifications. Briefly, 0.1 mL of the oil sample was added to 5 mL of thiobarbituric acid (TBA) solution, which was prepared by mixing 15 g of trichloroacetic acid (TCA), 0.375 g of TBA, and 2.1 g of hydrochloric acid (37% *w*/*w*). Samples were heated in a water bath at 95 °C for 10 min, then the samples were allowed to cool down to room temperature for 10 min, followed by centrifugation (Heraeus Multifuge 3SR Plus Centrifuge, Thermo Scientific Ltd., Loughborough, UK) at 10,000 g for 15 min. The absorbance of the supernatant was measured at 532 nm using a UV spectrophotometer (CECIL CE 1021 1000 Series, Cecil Instruments Ltd., Cambridge, UK). The concentrations of TBARS values were determined by using a standard curve prepared using a 1,1,3,3-tetraethoxypropane (TEP) standard curve (coefficient correlation (R^2^) = 0.9994). TEP standards between 0.01 to 0.20 µg/mL were prepared with trichloroacetic acid 7.5%. Measurements were conducted in triplicates for each emulsion batch.

### 2.11. Scanning Electron Microscopy (SEM)

SEM observations were performed according to the method described by Ding, Zhang, and Li [37] and Perone, et al. [52]. SEM was used to analyse the microstructure of HPMC-L and HPMC-H (4% *w*/*w*) gel samples and the emulsions, which were lyophilized in a vacuum freeze dryer (F.J. Stokes Corporation, Horsham, PA, USA) for three days and stored in a desiccator for 24 h. All the samples were cut into small slices and fixed on a brass holder. Then, the surface of the samples was coated with gold-palladium and observed in the microscope (FEI Quanta 600 FEG SEM version 2.4, FEI UK Ltd. Cambridge, UK) at an accelerating voltage of 20 kV and a magnification of 500×.

### 2.12. Statistical Analysis

One-way analysis of variance (ANOVA) was performed using SPSS version 25.0 (IBM SPSS Statistics for Windows, Armonk, NY, USA). Multiple pairwise comparisons using Tukey’s HSD test were used to compare the mean values (*p* < 0.05). A two-way ANOVA was conducted to evaluate the influence of two independent factors: the type of emulsion (CE-0, CE-2, CE-4, NE-0, and NE-2) and storage time (1, 7, and 14 days) on the creaming index and TBARS results.

## 3. Results and Discussion

### 3.1. Characteristic of HPMC Solutions

#### 3.1.1. Viscoelastic Properties of HPMC Solutions and Butter

The dynamic spectra of HPMC-L and HPMC-H solutions compared with butter are shown in Figure 1A,B. The mechanical strength of butter was greater than that of HPMC-L and HPMC-H solutions as illustrated by higher moduli values and a lower dependence on frequency. Butter showed a gel-like structure of a viscoelastic solid material as the G’ modulus was higher than G” modulus in the whole frequency range studied. In contrast, HPMC-L and HPMC-H showed a high frequency dependence and a predominant viscous over elastic behaviour at low frequencies (0.1–5 Hz) as expected from viscous fluids. At low temperatures (~20 °C) HPMC forms a solution in water as the water molecules form hydrogen bonds with the hydroxyl groups and form enclosed structures to surround the hydrophobic groups of the HPMC chain [27,36]. As frequency increased, G’ values increased more rapidly than G”, giving place to a moduli crossover point (5–100 Hz) that indicated a change from a liquid-like to a solid-like viscoelastic structure. Other authors have also observed a similar moduli cross over point when assessing the microstructure of HPMC concentration in mixed solutions with collagen [37]. At high frequencies the polymer chains did not have enough time to relax and detangle, so a fixed network was formed that stored the energy and behaved more like an elastic structure [37]. One could hypothesise that in the HPMC solutions under study, a higher number of hydrophilic–hydrophilic and hydrophobic–hydrophobic interactions formed and remained fixed at higher frequencies, giving place to an elastic HPMC network. In particular, at low temperatures, it has been reported that hydroxypropyl groups dominate the aggregation process [53].

Higher concentrations of both HPMCs gave place to an increase in G’ and G” moduli values and lower frequency dependence compared with lower concentrations of the polymers. These results could be due to the formation of a more structured solution as the concentration of the polymer increased. Other authors also found an increase in moduli values for the more concentrated HPMC solutions [29,30]. When comparing both types of HPMC polymers (-L and -H) at the same concentration of 4% *w*/*w*, the HPMC-L solution exhibited lower G’ and G” moduli values and a stronger dependence on frequency than the HPMC-H solution. These results indicate that HPMC-L had a weaker mechanical strength compared to an HPMC-H solution, and it could be explained by the lower amount of hydroxypropyl (7.2%) and methoxyl (21.4%) substitution levels in HPMC-L in comparison with HPMC-H (8.1% and 28.9%, respectively).

When hydroxyl groups in methylcellulose (MC) are substituted by hydrophobic groups, some hydrogen bonds within MC are prevented and the polymer derivate interacts more with the surrounding water [54]. Moreover, HPMC-H with a higher molecular weight range and a high ratio of methyl groups results in longer polymer chains with a high density of physical cross-linking sites, indicating a stronger network and more stable shape [55]. These properties could lead to a stronger and more stable HPMC-H solution.

#### 3.1.2. Temperature Ramp Viscoelasticity of HPMC Solutions and Butter

Figure 2 shows the gelation pattern as a function of temperature for HPMC-L and HPMC-H solutions. The viscoelastic behaviour of the HPMCs solutions during the heating process (Figure 2A,C) can be divided into three regions. At low and mild temperatures, from 20 °C to 30–40 °C, both HPMCs solutions showed a liquid-like viscoelastic behaviour, as G” values were higher than G’ values. As already discussed in Section 3.1.1. both HPMC polymers form an entangled polymer solution in water at low temperatures. In general, at this temperature range G’ and G” moduli values showed a smooth decrease as temperature increased. Then, as temperature increased, G’ and G” moduli increased dramatically and a cross-over point was observed. A similar temperature effect on HPMC solutions was reported by other authors [29,30,36,56]. The initial decrease in G’ and G” moduli values has been related to the disentanglement of cellulose chains, giving place to a loss of viscous and elastic consistency as temperature increased [36]. Then HPMC backbone flexibility and the dynamics of the water increased with temperature, unveiling the hydrophobic moieties of HPMC and giving place primarily to the formation of strong hydrophobic interactions in the formation of a three-dimensional network, with a contribution from interchain hydrogen bond formation [36,53,56]. This phase transition from a polymer solution to a gel is referred to as sol-gel transition and the temperature at which G’ increases abruptly is considered the starting point of the gelation temperature [30,36,54,57]. As the HPMC concentration of both HPMC types increased a decrease in the gelation temperature was observed. The lower onset temperature of aggregation and gelation of HPMC solutions has been attributed to higher hydrophobicity due to a higher concentration of methoxy and hydroxypropyl in the solution [26]. As explained before, the hydrophobic interactions among HPMC chains are the principal forces for the formation of a three-dimensional network. However, when comparing the thermogelation of both types of HPMC at the same concentration at 4%, the hydrophobicity did not define the changes in gelation temperatures. The lower gelation temperature of HPMC-L could be explained due to its lower molecular weight which allowed easier interactions between hydrophobic groups for the formation of the gel. These results agreed with previous observations on the effect of temperature on gelation of HPMC with different molecular weights [28]; low molecular weight HPMC polymers presented a rod-like structure, which allowed a parallel arrangement along the chain leading to an easier aggregation in comparison with higher molecular weight HPLC which had a random coil configuration [28].

On cooling, a two-stage process was observed (Figure 2B,D): first a gradual decrease, and second, a dramatic decrease in G’ and G” moduli values. A similar degelation process has been described previously where initially, hydrophobic associations were gradually weakened, the network dissociated, and then intermolecular hydrogen bonding and water cages reformed around the hydrophobic moieties [36]. HPLC-H gels showed a more pronounced decrease in the G’ and G” values than HPMC-L. These results indicated that HPMC-L gels presented higher mechanical strength during the cooling process. HPMC-L could have a higher potential to be applied as a structuring agent in complex emulsions, as HPMC-L gel could retain a liquid phase and provide better stability due to its strength after heating. These results are crucial for the development of complex systems such as emulsions, in which liquid components (oils) must be retained in the gel network over a series of thermal cycles (e.g., baking).

#### 3.1.3. Firmness and Spreadability of HPMC Solutions

The firmness and spreadability properties of a colloid define how it will behave during certain unit operations (mixing, sheeting, cutting); thus, these parameters will give us an indication of the technological functionalities of the HPMC solutions. The firmness and spreadability of HPMC solutions were analysed and compared with butter in order to select the suitable type and concentration of HPMC that could be used to formulate emulsions that could be used as a saturated fat replacer. As shown in Table 2, there were no significant differences in both firmness and work of shear between butter and 4% HPMC-L solution. These results suggest that HPMC-L 4% solution will behave more like butter than the other solutions during food production operations such as mixing or dough sheeting. The concentration of HPMC had a significant effect (*p* < 0.05) on the firmness and spreadability of HPMC solutions. It was found that firmness and work of shear of HPMC solutions increased significantly (*p* < 0.05) with increasing concentration; high values of work of shear suggest a higher force is needed to induce shearing deformation, which indicates low spreadability. An increase in HPMC concentrations could lead to an increase in cellulose hydrophobicity and the formation of a transient three-dimensional network [58,59]. Regarding the effect of HPMC type, at the same concentration of 4% *w*/*w*, the HPMC-H solution presented significantly (*p* < 0.05) higher values of firmness and work of shear than the HPMC-L solution. These results could be explained by the higher mechanical strength of the HPMC-H solution, at room temperature, as explained in Section 3.1.1.

#### 3.1.4. Scanning Electron Microscopy of HPMC Solutions

Scanning electron microscopy (SEM) was used to further understand the microstructure of HPMC gels at 4% *w*/*w* as shown in Figure 3. The pores of HPMC-L gel were smaller than those of HPMC-H. These results could be related to differences in the polymer configuration. Higher molecular weight HPMC presents a random-coil configuration leading to a slower gel network formation, and thus potentially to the formation of larger cells [28], creating a more open structure of hydrophobic association between HPMC chains. In comparison, HPMC-L with lower molecular weight and a rod-like structure [28] could more easily form hydrophobic interactions, giving place to a more regular and compact network.

The molecular structure of HPMC and its concentration in solution were determinant factors for the behaviour of the polymer in solution during heating, specifically in the mechanical strength of the structures and the onset of the gelation temperature. HPMC-L had a higher surface activity than HPMC-H (Appendix A) and its lower molecular weight gave place to thermo-reversible gels that presented higher stability during cooling and a stronger structure. Therefore, HPMC-L was selected as the hydrocolloid for incorporation in complex emulsions.

### 3.2. Conventional Emulsion and Nanoemulsion Stabilized by Lecithin and HPMC-L

#### 3.2.1. Viscoelastic Properties of Emulsions

The dynamic spectra of CE and NE stabilised with lecithin and HPMC in comparison to butter are shown in Figure 4. All samples showed a solid-like viscoelastic behaviour as the G’ modulus was higher than the G” modulus in the whole frequency range studied. The mechanical strength of butter was greater than that of CE and NE samples as illustrated by higher moduli values and lower dependence on frequency. In comparison, HPMC solutions studied in Section 3.1.1 showed a fluid viscoelastic behaviour at low temperatures because HPMC forms a solution in water as the water molecules form hydrogen bonds with the hydroxyl groups and form enclosed structures to surround the hydrophobic groups of the HPMC chain [27,36].

Emulsions with higher concentrations of HPMC (CE-4) showed higher G’ and G” moduli values and lower frequency dependence compared to CE-2, CE-0, NE-0, and NE-2 (Figure 4). These results could be explained by the increase in the polymer concentration, which leads to the formation of a higher number of entanglements in the network and thus to a more structured solution. This finding was in agreement with other authors that found an increase in moduli values at higher concentrations of HPMC in the solution [29,30]. As frequency increases, moduli values tend to approach. This could be explained by a reduction in the strain response of the sample, as at higher frequencies the time of strain response is shorter and samples may not be able to follow the stress variation.

#### 3.2.2. Temperature-Dependent Rheological Behaviour of Emulsions

As discussed in Section 3.1.2, aqueous solutions of HPMC-L undergo thermoreversible sol–gel transitions [36]. The gelation as a function of the temperature of CE and NE stabilised by lecithin and HPMC-L (0, 2, or 4%) was investigated to evaluate the effect of the o/w emulsion on the gelation behaviour of HPMC (Figure 5). At low temperatures (20 °C–40 °C) both moduli values showed a plateau or a smooth decrease in moduli values. These results could be related to a sort of thermal softening as the intermolecular hydrogen bonds are gradually weakened by the increase in temperature [36]. The dynamic spectra of both CE-0 and NE-0 differed significantly and moduli values were smaller due to the absence of HPMC in their formulation. As the temperature increased, G’ and G” moduli increased abruptly, marking the starting point of the gelation temperature [30,54,57]. The gelation temperature of the emulsions was influenced by HPMC concentration following the same trend observed for HPMC solutions (Figure 2; as HPMC concentration increased, the emulsion gelation temperature decreased). As discussed in Section 3.1.2, these results could be due to a higher hydrophobicity from methoxy and hydroxypropyl groups when increasing HPMC content, leading to a lower onset temperature of aggregation and gelation of HPMC solutions [26]. The thermorheogram shows a fluctuation of moduli values after the gelation temperature; G’ and G” values decreased. This change in the viscoelastic behaviour during heating could be due to phase separation between HPMC gel and the liquid components after the start of the gelation process. Other authors observed that phase separation is caused by a hydrophobic association between the HPMC chains when the temperature increases [26,29].

During the cooling cycle, from 90 °C–50 °C (Figure 5B), the moduli values of the emulsions showed a smooth increase; the highest G’ values were at around 50 °C for all samples, which was a similar temperature to the gelation temperature during the heating cycle. Then the G’ values showed a smooth decrease until the end of the cooling cycle (20 °C). This behaviour could be related to the effect that high temperatures have on the formation and stability of hydrophobic interactions and the stability of the gel network. As soon as the temperatures were lower than the gelation temperature (<50 °C), emulsion moduli values showed a gradual decrease. These observations differ from the degelation behaviour of HPMC solutions discussed in Section 3.1.2. In the case of HPMC solutions, a two-stage process was observed: initially, cooling hydrophobic associations were gradually weakened and the network dissociated, giving place to a smooth decrease in moduli values; then, water caged around hydrophobic moieties and intermolecular hydrogen bonds were formed producing a dramatic decrease on moduli values. In this study, the interaction between the emulsifiers, as explained above, may have increased the moduli values at the start of the degelation process, until hydrophobic interactions are weakened and hydrophilic ones prevail.

#### 3.2.3. Firmness and Spreadability of Complex Conventional Emulsions and Nanoemulsions

The evaluation of textural properties such as firmness and spreadability was performed to gather an indication of the behaviour of the samples in food processing operations such as scooping, mixing, or sheeting where penetration and shear forces are applied to the food products. As shown in Table 3, The firmness and spreadability values of butter and NE-2 were similar (*p* > 0.05), suggesting that when NE-2 is used in mixing or dough sheeting operations, it will respond to the forces in a similar way to butter. The concentration of HPMC had a significant effect (*p* < 0.05) on the firmness and spreadability of emulsions (Table 3). A higher HPMC concentration in CE required significantly higher (*p* > 0.05) penetration and shear forces to induce a deformation in the samples, thus CE-4 was firmer and more difficult to spread than CE-2. These results agree with the viscoelastic behaviour of the samples, where it was also observed that higher HPMC concentration gave place to more structured samples (Section 3.2.1). As discussed previously, an increase in HPMC concentration could lead to an increase in hydrophilic–hydrophilic interactions that dominate the formation of the HPMC network structure at room temperature [36,56]. In fact, polysaccharides are often added to oil-in-water emulsions to increase the continuous phase viscosity, giving place to advantageous textural properties and better physical stability of the emulsion [60].

When comparing CE-2 and NE-2 with the same HPMC concentration (2%), NE presented significantly (*p* < 0.05) lower values of firmness and shear (higher spreadability) than CE. Although this trend was not observed in the analysis of the viscoelastic behaviour of the samples (Figure 4, the dynamic spectra showed that CE-2 and NE-2 had similar moduli values), previous studies on the structure of nanocellulose materials reported that the network structure between cellulose chains formed by intermolecular and intramolecular hydrogen bonds were destructed by the shearing forces during HPH [61]. In contrast with results reported previously [61], no effect of the HPH process was observed in the emulsion microstructural behaviour. Higher deformation forces applied in the penetration and spreadability tests may have also contributed to the breakage of parts of the network that were affected by the HPH treatment and through these measurements, the changes in the microstructure of the samples were detected.

#### 3.2.4. Scanning Electron Microscopy of Complex Conventional Emulsion and Nanoemulsion

SEM images of emulsions are shown in Figure 6. The oil droplets (white arrows) were covered by an HPMC layer. As expected, nanoemulsions (NE-2) showed a distribution of smaller oil droplets than CE-2. Moreover, when comparing these two samples, at the same HPMC concentration (2%) it could be observed that the HPMC network structure was discontinuous in NE-2 in comparison to the continuous layer in CE-2. This result could be explained by the destruction of HPMC during the high-pressure homogenisation process. There were high shear forces and cavitation during high-pressure homogenization [62,63], where the high intensity of the disruptive forces could break up the oil droplets into smaller size droplets and destroy the HPMC network [64,65].

#### 3.2.5. Physical and Chemical Stability of Emulsions

##### Creaming in Emulsions

There were significant interactions (*p* < 0.05) between the emulsion type and storage time when the CI of the emulsions was evaluated (Figure 7). The effect of a polysaccharide on an emulsion’s physical stability depends on the polysaccharide’s molecular structure and concentration [60]. In our study, the concentration of HPMC played an important role in CI values. At both storage temperatures, CE-0 and NE-0 had a significantly (*p* < 0.05) higher CI value than CE-2, CE-4, and NE-2. These results could be attributed to the effect of HPMC in the mechanical properties of the emulsions, as higher concentrations of HPMC gave place to more structured and firmer viscoelastic gels (Figure 4 and Table 3) that could have prevented droplet aggregation more effectively than less structured emulsions. Some polysaccharides at certain concentrations form entangled networks that trap the droplets and inhibit their movement [66]. These results were in agreement with previous research where it was observed that the creaming index decreased with increasing viscosity of the continuous phase of o/w emulsions stabilised with HPMC and sodium dodecylsulfate (SDS) [67]. Borreani, et al. [68] also observed that o/w emulsions stabilised with cellulose ether did not present phase separation during a 3-day storage period. Moreover, the use of mixed emulsifiers could have a dual stabilisation mechanism: the aforementioned thickening effect of the continuous phase by HPMC and the formation of lecithin-HPMC complexes, forming cross-links around the hydrophobic sites of HPMC [55], may have improved the stability of the emulsion. The stability of o/w emulsions prepared with hydrophobically-modified hydroxyethyl cellulose (HMHEC) was based on (i) the thickening effect caused by the alkali chains of the HMHEC and (ii) the adsorption of the HMHEC at the interface forming a film that prevented coalescence [59].

In general, an increase in storage time from day 1 to day 7 gave place to a significant increase (*p* < 0.05) in CI values for CE-0 and NE-0. Oil droplet aggregation including flocculation and coalescence take place over time, leading to the formation of a creaming layer in emulsions [23]. When evaluating the effect of the droplet size on emulsion stability it was observed that NE-0 presented higher stability, with significantly lower CI values (*p* < 0.05) than CE-0, which could be due to the smaller droplets size (MDD) and narrower size distribution values (PDI) of the NE (Appendix A), which could be explained by the effect of high-pressure during emulsion formation; an increase in shear forces and cavitation during high-pressure homogenisation resulted in particle size and polydispersity reduction [62,63]. The CI values of emulsions can be influenced by various factors including droplet size and density of the dispersed (oil) phase. Stokes’ law defines the rate of gravitational separation: it can be decreased by a decrease in droplet size, a decrease in density difference between the dispersed phase and continuous phase, and an increase in the viscosity of the continuous phase [23]. A decrease in particle droplet size gave place to a decrease in the attractive forces between the droplets [62,69]; smaller droplets have better stability against droplet coalescence and flocculation because of the reduction in Brownian motion and gravitation forces [69], allowing nanoemulsions to be protected against flocculation phenomena.

##### Lipid Oxidation in the Emulsions over Storage Time

The TBARS method was used to measure the secondary products of lipid oxidation in the emulsions over storage time (1, 7, and 14 days) at two temperatures (4 °C and 20 °C). There were no significant interactions (*p* > 0.05) between the type of emulsions and storage time at any of the two storage temperatures tested. The mean plots are shown in Figure 8 (A and B, respectively). HPMC concentration improved the lipid oxidative stability of the emulsions; CE-2, CE-4, and NE-2 showed significantly lower (*p* < 0.05) TBARS values than CE-0 and NE-0 (Figure 8A). In o/w emulsions, oxidation is usually initiated in the aqueous phase where the concentration of prooxidants is higher than in the oil droplets [24]. HPMC may have a dual function in the improvement of the lipid oxidation stability of emulsions: (i) in the formation of a protective interfacial layer with lecithin, as HPMC adsorption in the interface of the oil droplets will increase the thickness of this physical layer, thus improving the protection of the lipids and avoiding prooxidant substances to come into contact with them [23,24]; (ii) as a thickener, limiting the diffusion of prooxidants [68].

Complex nanoemulsions did not show significantly different TBARS values than conventional emulsions (Figure 8); these results meant that droplet size had no effect on TBARS values of emulsions stabilised with HPMC. In contrast, it has been reported in previous literature that a smaller droplet size could lead to higher lipid oxidation of nanoemulsions due to an increased droplet surface area [3,70]. In this study, the stabilisation functionality exerted by HPMC had a greater and limiting effect on the lipid oxidation rate than the increase in droplet surface area. The viscoelastic film of absorbed HPMC on the droplet surfaces protected oil droplets against prooxidant substances. This result was in agreement with another author [25], who reported that oil droplet size of oil-in-water emulsions stabilised by sodium caseinate (CAS) had no significant influence on the rate of lipid oxidation, whereas the ability of CAS to form a thick interfacial layer surrounding droplets and scavenge free radicals has been proven to be effective in inhibiting lipid oxidation [25].

The changes in the TBARS values of emulsions through storage time (Figure 8B) were significant (*p* < 0.05). TBARS values increased when storage time increased from day 7 to day 14 at 4 °C and from day 1 to day 7 at 20 °C. There are many factors that could affect the oxidation reaction of emulsions such as the chemical structure of lipids, oxygen concentration, temperature, antioxidants, droplet characteristics, and light [3,71,72]. Therefore, longer storage time gave place to a prolonged exposition to increased oxygen, light, and temperature, increasing TBARS on emulsions.

## 4. Conclusions

This study showed that the molecular structure of HPMC and its concentration in solution were determinant factors for the behaviour of the polymer in solution during heating, specifically in the mechanical strength of the structures and the onset of the gelation temperature. A higher concentration of hydrophobic groups gave place to HPMC solutions with higher mechanical strength. However, lower molecular weight HPMC formed thermoreversible gels that presented higher stability during cooling and higher final strength. Moreover, this study revealed that the combination of emulsifiers (lecithin and HPMC) had a positive effect on the rheological and texture properties and on the physical and chemical stability of conventional emulsions and nanoemulsions. The results demonstrated that a combination of lecithin and HPMC improved the physical and lipid oxidative stability of the emulsions during storage, due to (i) the formation of a thicker protective layer of lecithin and HPMC adsorbed in the interface of the oil droplets and (ii) the formation of a stronger network in the continuous phase. Nanoemulsions showed higher spreadability than conventional emulsions due to the disruptive forces of high-pressure homogenization on the emulsion structure. A nanoemulsion formulated with 5% lecithin and 2% HPMC could be used as a suitable saturated fat replacer on bakery products, as it showed a similar firmness and spreadability to butter, indicating a similar deformation response in processing operations. These results will be the basis to formulate saturated fat replacers with specific technological functionalities to achieve the right performance in food manufacturing operations such as mixing, dough sheeting, baking, and storage.

## Figures and Tables

**Figure 1 foods-11-02536-f001:**
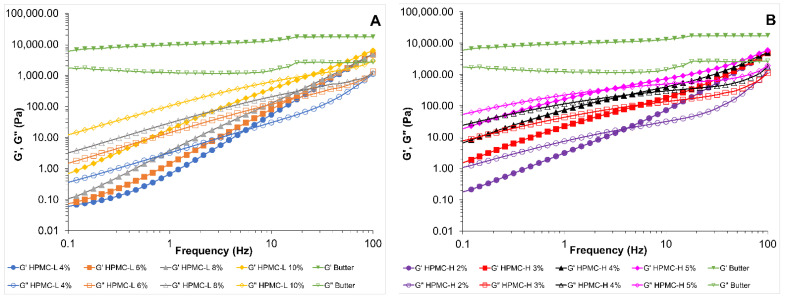
G’ and G” moduli of HPMC solutions and butter as a function of frequency. (**A**) HPMC-L solutions at different concentrations (4% blue circles; 6% orange squares; 8% grey triangles; 10% yellow diamond) and butter (green down-pointing triangle). (**B**) HPMC-H solutions at different concentrations (2% purple circles; 3% red squares; 4% pink triangles; 5% black diamond) and butter (green down-pointing triangle). Filled symbols correspond to storage modulus (G’) and open symbols to loss modulus (G”).

**Figure 2 foods-11-02536-f002:**
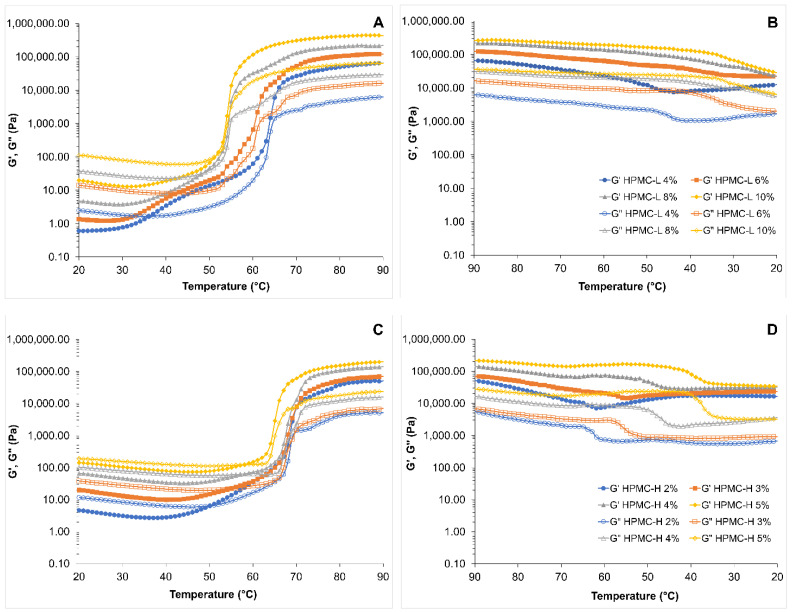
G’ and G” moduli of HPMC solutions as a function of temperature. (**A**) temperature ramp up for HPMC-L; (**B**) temperature ramp down bot HPLC-L; (**C**) temperature ramp up for HPMC-H; (**D**) temperature ramp down bot HPLC-H. Filled symbols correspond to elastic modulus (G’) and open symbols to viscous modulus (G”). Frequency = 1 Hz, heating rate = 1 °C/min.

**Figure 3 foods-11-02536-f003:**
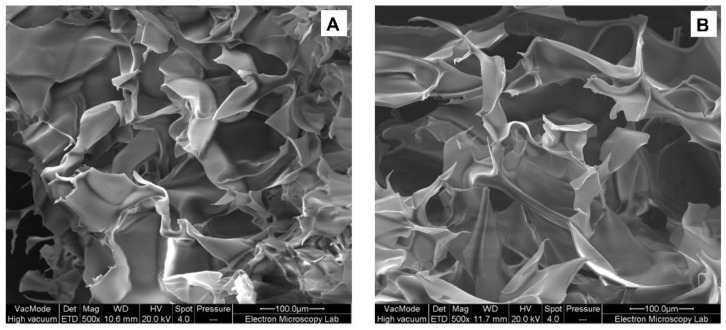
SEM images of: (**A**) HPMC-L gel, (**B**) HPMC-H gel at 500×. Micrographs were obtained under vacuum, no pressure was recorded (---).

**Figure 4 foods-11-02536-f004:**
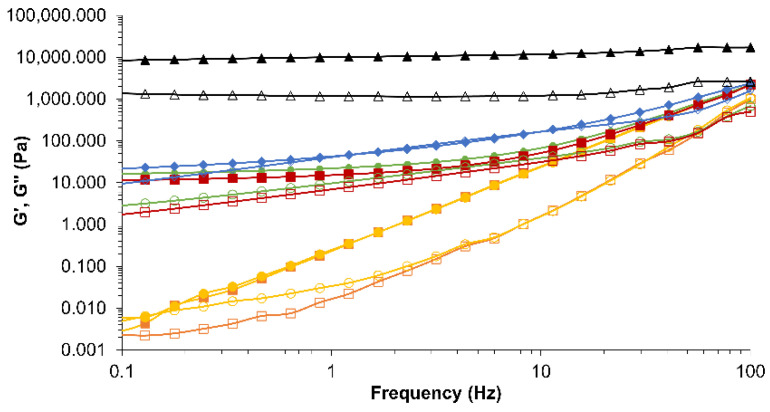
Dynamic moduli as a function of frequency of emulsions and butter 
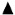
. Conventional emulsion (CE-0 
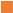
; CE-2 
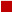
; CE-4 
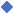
). Nanoemulsion (NE-0 
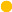
; NE-2 
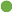
). Filled symbols correspond to elastic modulus (G’) and open symbols to viscous modulus (G”).

**Figure 5 foods-11-02536-f005:**
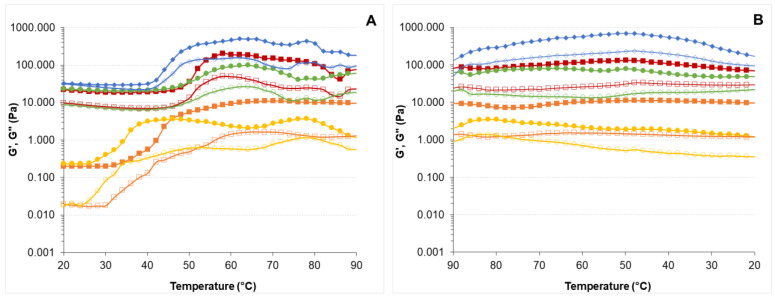
G’ and G” moduli of complex emulsions as a function of temperature. (**A**) Temperatur ramp up; (**B**) Temperature ramp down. Conventional emulsion (CE-0 
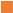
, CE-2 
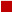
; CE-4 
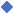
). Nanoemulsion (NE-0 
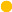
, NE-2 
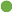
). Filled symbols correspond to elastic modulus (G’) and open symbols to viscous modulus (G”). Frequency = 1 Hz, heating rate = 1 °C/min.

**Figure 6 foods-11-02536-f006:**
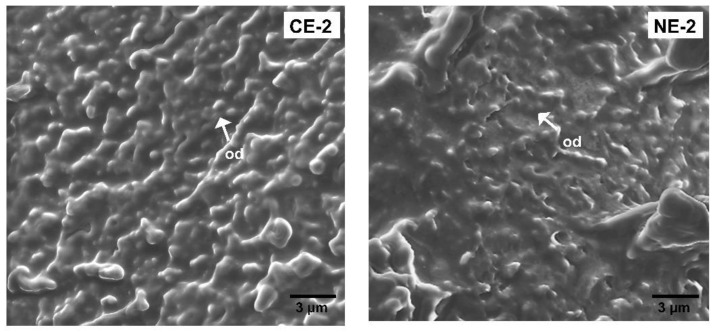
SEM images of conventional emulsions (CE) and nanoemulsions (NE) stabilised with lecithin (5%) and HPMC (2%). 500×. od: oil droplets.

**Figure 7 foods-11-02536-f007:**
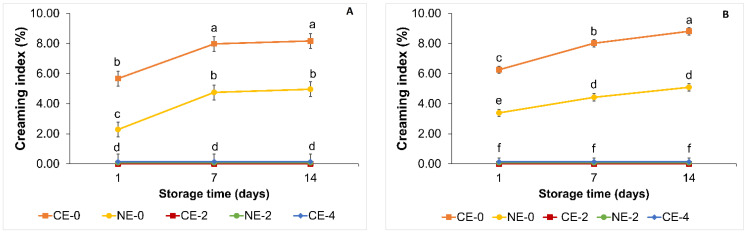
Creaming index of emulsions during storage (day 1, 7, and 14) at different storage times: (**A**) 4 °C and (**B**) 20 °C (mean values and 95% HSD confidence intervals). Conventional emulsion (CE-0 
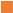
, CE-2 
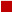
; CE-4 
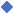
). Nanoemulsion (NE-0 
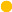
; NE-2 
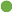
). Values with different superscript letters are significantly different (*p* < 0.05).

**Figure 8 foods-11-02536-f008:**
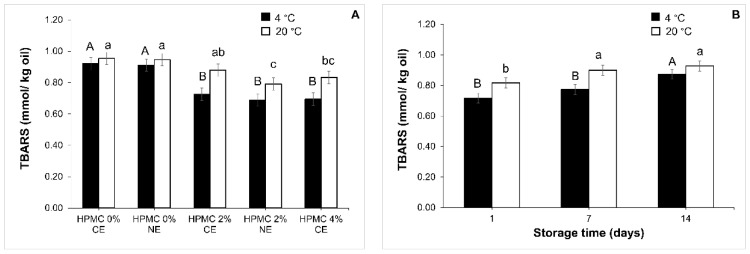
Mean values and 95% HSD confidence intervals. (**A**): TBARS mean values according to the type of in emulsions at 4 °C (filled bars) and 20 °C (empty bars), conventional emulsions (CE-) and nanoemulsion (NE-) stabilised with lecithin and HPMC (0, 2 and 4%); (**B**): TBARS mean values according to the storage day at 4 °C and 20 °C. Different capital letters indicate significant differences in the mean at 4 °C and different lower-case letters indicate significant differences in the mean at 20 °C (*p* < 0.05).

**Table 1 foods-11-02536-t001:** Properties of the HPMC used, as provided by the manufacturer.

Samples	Viscosity (mPa·s)	Methoxy (%)	Hydroxypropyl (%)	Molecular Weight (g/mol)
HPMC-L	80–120	21.4	7.2	26,000
HPMC-H	3000–56,000	28.9	8.1	86,000

**Table 2 foods-11-02536-t002:** Firmness and work of shear of different concentrations of HPMC solutions and butter.

Samples	Firmness (N)	Work of Shear (N·s)
Butter	0.35 ^f^ (0.06)	0.26 ^de^ (0.04)
HPMC-L 4%	0.24 ^f^ (0.02)	0.10 ^e^ (0.01)
HPMC-L 6%	0.92 ^e^ (0.02)	0.48 ^d^ (0.02)
HPMC-L 8%	2.39 ^d^ (0.06)	1.31 ^c^ (0.10)
HPMC-L 10%	5.17 ^b^ (0.46)	2.86 ^b^ (0.26)
HPMC-H 2%	0.87 ^e^ (0.06)	0.49 ^d^ (0.03)
HPMC-H 3%	2.23 ^d^ (0.14)	1.39 ^c^ (0.09)
HPMC-H 4%	3.44 ^c^ (0.22)	2.94 ^b^ (0.22)
HPMC-H 5%	8.05 ^a^ (0.26)	5.77 ^a^ (0.19)

Indicated values are reported as means (standard deviation). Values with the different superscript letters are significantly different (*p* < 0.05) between samples in the same column.

**Table 3 foods-11-02536-t003:** Firmness and work of shear of butter, conventional emulsions (CE) and nanoemulsion (NE) stabilised with HPMC (0%, 2% and 4%).

Samples	Firmness (N)	Work of Shear (N·s)
Butter	0.33 ^c^ (0.03)	0.24 ^bc^ (0.06)
CE-2	0.44 ^b^ (0.02)	0.29 ^b^ (0.01)
NE-2	0.30 ^c^ (0.02)	0.18 ^c^ (0.02)
CE-4	1.87 ^a^ (0.11)	1.14 ^a^ (0.06)

Indicated values are reported as means (standard deviation). Values with the different superscript letters are significantly different (*p* < 0.05) between samples in the same column.

## Data Availability

The data presented in this paper are openly available in the University of Reading Research Data Archive at https://doi.org/10.17864/1947.000410 (accessed on 16 August 2022).

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
