# Peer review of "Development of Saturated Fat Replacers: Conventional and Nano-Emulsions Stabilised by Lecithin and Hydroxylpropyl Methylcellulose"

_foods, 2022, doi:10.3390/foods11162536_

Round 1
Reviewer 1 Report
The manuscript titled " Development of saturated fat replacers: nanoemulsion stabilised by lecithin and hydroxylpropyl methylcellulose" is quite interesting. The experimental plan is well executed. However, it still needs some revisions.
1. In line 62-65, please rephrase this sentence. (sentence begins with “However, the combination of…”)
2. How the concentrations of HPMC-L and HPMC-H were determined, and why more HPMC samples of the same concentration were not prepared for comparison?
3. In line 122, please delete this sentence (sentence begins with “HPMC-L was selected due to…”), and supplemented to the explanation that this conclusion has been obtained through data analysis. Does “HPMC-L was selected due to forming a more stable gel network than HPMC-H.” contradict the statement “These properties could lead to a stronger and more stable HPMC-H solution” in line 277?
4. In line145, “at 0.00, 0.001, 0.05, 0.1, 0.1, 0.3, 0.5, 0.8 and 1.0” is wrong.
5. In line156, why are the temperature settings of different HPMC solutions, emulsions and butter different in amplitude sweep and frequency sweep measurements? (“Amplitude sweep, frequency sweep measurements were performed at 20 °C for HPMC solutions and emulsions, and at 30 °C for butter.”)
6. The data in Table 2 do not indicate “firmness and work of shear of HPMC solutions increased significantly with increasing concentration” and “at the same concentration of 4% w/w, HPMC-H solution 354 presented significantly higher values of firmness and work of shear than HPMC-L solution”.
7. Why the data of CE-0 and NE-0 are not shown in sections 3.2.2 and 3.2.3?
8. In line 328, how to prove that “HPMC-L gel could retain a liquid phase”?
9. Why is the work of shear of the butter in Table 2 and Table 3 so different?
10. Please revise HMPC to HPMC.
11. There is an error with punctuation marks in line 466
12. In section 3.2, most of the data was detected in liquid state, why choose HPMC-L, which can form a better gel network?
13. There are some formats in the reference that need to be adjusted to unity, such as the initial letters of all words in the paper name.
Author Response
- In line 62-65, please rephrase this sentence. (sentence begins with “However, the combination of…”)
The sentence has been rephrased (L63-67).
- How the concentrations of HPMC-L and HPMC-H were determined, and why more HPMC samples of the same concentration were not prepared for comparison?
Preliminary work was carried out to select the concentrations of HMPC-L and HPMC-H to study and compare for this research project. Initially, solutions of both HMPC types were prepared at 4, 6, 8 and 10% w/w. However, HMPC-H dispersion at 6, 8 and 10% was very difficult and the final solutions presented significantly higher moduli values than all the other solutions. Further work was carried out to evaluate the viscoelasticity of HMPC-H solutions at 2, 3, 4 and 5% w/w. These solutions presented moduli values in a similar range to HMPC-L solutions at 4, 6, 8 and 10% (storage modulus was ~4,000 – 6,000 Pa and loss modulus was ~1,000 – 2,000 Pa at strain amplitude of 1.0% with frequency of 1 Hz). Since, both HMPC types showed a very different rheological behaviour, it was very difficult to select further concentration for comparison, and the authors decided to study each HMPC separately first, and then compare them at one single concentration as the number of samples to prepare was already high (8 samples in total). A sentence has been added in the manuscript to clarify the selection of concentrations of the HMPC solutions (L108-109).
- In line 122, please delete this sentence (sentence begins with “HPMC-L was selected due to…”), and supplemented to the explanation that this conclusion has been obtained through data analysis. Does “HPMC-L was selected due to forming a more stable gel network than HPMC-H.” contradict the statement “These properties could lead to a stronger and more stable HPMC-H solution” in line 277?
The sentence has been modified (L126-131) to reflect that HMPC selection was based on data analysis as discussed in section 3.1.2. ‘These results indicated that HPMC-L gels presented higher mechanical strength during the cooling process. HPMC-L could have higher potential to be applied as structuring agent in complex emulsions, as HPMC-L gel could retain a liquid phase and provide a better stability do to its strength after heating. (L335-339)’.
- In line145, “at 0.00, 0.001, 0.05, 0.1, 0.1, 0.3, 0.5, 0.8 and 1.0” is wrong.
It has been corrected (L153).
- In line156, why are the temperature settings of different HPMC solutions, emulsions and butter different in amplitude sweep and frequency sweep measurements? (“Amplitude sweep, frequency sweep measurements were performed at 20 °C for HPMC solutions and emulsions, and at 30 °C for butter.”)
Authors decided to use different temperatures to assess the viscoelastic properties of HPMC solutions and emulsions (20 °C) in comparison to butter (30 °C) based on previous experience working with biscuit industrial production processes. When butter or shortenings (palm fat, etc.) are added in the biscuit dough mixing process they are usually tempered at 25-30 °C before adding them to the mixture. This was explained in Lines 163-166. Authors suggested that HMPC solutions and emulsions could be used at room temperature (20 °C) for the mixing process. Higher temperatures as (20 – 40 °C) gave place to a smooth decrease in the viscoelastic properties of the emulsions, which could have implications on the oil structuring stability and oil leaking into the dough during mixing and before the thermal gelation happens during baking, reducing the functionality of the emulsions in bakery systems.
- The data in Table 2 do not indicate “firmness and work of shear of HPMC solutions increased significantly with increasing concentration” and “at the same concentration of 4% w/w, HPMC-H solution 354 presented significantly higher values of firmness and work of shear than HPMC-L solution”.
Values of work of shear in Table 2 were wrong and have now been rectify (L368). After this correction, the discussion of firmness and work of shear of the HMPC solutions in section 3.1.3. follows the trend presented in Table 2 (L359-367).
- Why the data of CE-0 and NE-0 are not shown in sections 3.2.2 and 3.2.3?
Viscoelasticity (section 3.2.2) of CE-0 and NE-0 was also evaluated and has been added in an updated Figure 5-R1 (see below). The dynamic spectra of both CE-0 and NE-0 differed significantly and moduli values were smaller due to the absence of HPMC in their formulation (L421-423).
Firmness and work of shear (section 3.2.3), of CE-0 and NE-0 could not be measured using a TTC spreadability rig. When using the spreadability rig, the material is set in the lower cone in a texture analyser the upper probe goes down and fits perfectly within the lower probe. The important action that the test is designed to measure, spreadability, occurs only in the later stages of the test. During these stages the product is squeezed out from between the male and female cones. The cone-shaped holder offers no locations into which the product can be packed or compressed, so the product flows outward between the male and female cone surfaces. However, enough sample should remain in the lower cone to acquire accurate results (Figure A). However, in this study most of the CE-0 and NE-0 samples flowed out the lower cone when they were penetrated (Figure B below) giving inaccurate values of firmness and work of shear.
Figure A) Spreadability test using the TTC probe (https://texturetechnologies.com/application-studies/spreadability). B) Wrong performance of the spreadability probe when the sample has very low viscosity and flows out of the lower cone.
Figure 5-R1. G' and G" moduli of complex emulsions as a function of temperature. Conventional emulsion (CE-0 ; CE-2 ; CE-4 ). Nanoemulsion (NE-0 ; NE-2 ). Filled symbols correspond to elastic modulus (G') and open symbols to viscous modulus (G"). Frequency = 1 Hz, heating rate = 1 °C/min.
- In line 328, how to prove that “HPMC-L gel could retain a liquid phase”?
Authors hypothesised that HPMC-L gel could retain liquid oil in its network better than HMPC-H, as the viscoelastic moduli of HMPC-L showed higher values than HMPC-H after cooling. However, the oil phase retention capacity of the two HMPC gels was not proved at this stage. The aim of this work was to preselect the best HPMC to formulate and produce complex nanoemulsions that could mimic the mechanical properties of butter as close as possible. Once, this objective was achieved, authors worked on a following research project on the application of the complex nanoemulsions in bakery systems (biscuits) and the oil retention in dough and final biscuit was assessed. This work will be submitted for publication in the coming months.
- Why is the work of shear of the butter in Table 2 and Table 3 so different?
Table 2 has been corrected (L368).
- Please revise HMPC to HPMC.
HMPC has been replaced by HPMC through the manuscript.
- There is an error with punctuation marks in line 466
The error has been corrected (L480).
- In section 3.2, most of the data was detected in liquid state, why choose HPMC-L, which can form a better gel network?
When HPMC solutions were studied in section 3.1. a fluid viscoelastic behaviour (at room temperature) was observed. HPMC formed a solution in water as the water molecules formed hydrogen bonds with the hydroxyl groups and formed enclosed structures to surround the hydrophobic groups of the HPMC chain. HPMC-L solutions had a higher surface activity than HPMC-H solutions (Figure S1) and its lower molecular weight gave place to thermo-reversible gels that presented higher stability during cooling and a stronger structure after cooling. Therefore, HMPC-L was selected for formulating and studying conventional and nanoemulsions stabilised by HPMC and lecithin (section 3.2). Emulsion samples showed a solid-like viscoelastic behaviour as the G' modulus was higher than G" modulus in the whole frequency range studied.
- There are some formats in the reference that need to be adjusted to unity, such as the initial letters of all words in the paper name.
The format of the references has been adjusted following reviewer’s suggestion.
Reviewer 2 Report
1. The strain in the frequency scan test is not reflected, and it is recommended to add experimental data from the amplitude sweep.
2. In the analysis of the results of the frequency sweep, such as line 265-268, "When comparing both types of HMPC polymers (-L and -H) at the same concentration of 4% w/w, HPMC-L solution exhibited lower G' and G" moduli values, a stronger dependence on frequency than HPMC-H solution.” Is modulus size used here to compare the strength of frequency dependence? Is there an inevitable correlation between frequency dependence and modulus size? Are there any corresponding references? The article has several similar explanations and the same questions as here.
Author Response
- The strain in the frequency scan test is not reflected, and it is recommended to add experimental data from the amplitude sweep.
The strain amplitude used in the frequency sweeps have been added in the manuscript: 1.0 % for HPMC solutions and emulsions, and 0.01 % for butter were selected for the frequency scan test (L176). Results of the amplitude sweeps of HPMC solutions (Figure S2) and emulsions (Figure S3) have been added as supplementary materials.
- In the analysis of the results of the frequency sweep, such as line 265-268, "When comparing both types of HMPC polymers (-L and -H) at the same concentration of 4% w/w, HPMC-L solution exhibited lower G' and G" moduli values, a stronger dependence on frequency than HPMC-H solution.” Is modulus size used here to compare the strength of frequency dependence? Is there an inevitable correlation between frequency dependence and modulus size? Are there any corresponding references? The article has several similar explanations and the same questions as here.
No, moduli size was not used to compare frequency dependence and there was no significant correlation between these two aspects discussed from the dynamic spectra presented. The values of the moduli were used to discuss the type of behaviour (solid viscoelastic or liquid viscoelastic) and the magnitude of the moduli was evaluated to assess which samples were firmer than the others. However, the frequency dependence for the moduli was assessed through the observation of the slope of the curves. For both HPMCs the moduli values increased when increasing the frequency; however, HPMC-L showed higher frequency dependence. Frequency dependence of a sample is related to the microstructure of the sample.
Reviewer 3 Report
This manuscript has investigated the influence of hydroxylpropyl methylcellulose type and concentration on the physicochemical properties of conventional emulsions and nanoemulsions. The topic is of great importance and the results are interesting and useful. However, there are some concerns that should be addressed:
The title of the manuscript is not appropriate. You have compared conventional emulsions with nanoemulsions but only nanoemulsion is written in the title.
Line 87: Please add the type of oil you used in this study
Line 99: delete “and the two types of HPMC (Biosynth carbosynth Limited, UK)” it was written in line 92.
Line 112: How the HPMC solution was prepared and when it was added to emulsions?
Lines 121-127: Paraphrase and delete the repeated sentences.
Line 136: ... were left for …
Line 241: are
Line 281: G' and G" moduli of HPMC solutions and butter as a function of frequency
Table 2: The statistical analysis for “Work of shear” is not correct.
Line 575: (0, 2 and 4 %)
Author Response
- The title of the manuscript is not appropriate. You have compared conventional emulsions with nanoemulsions but only nanoemulsion is written in the title.
The title has been modified following reviewer’s suggestions.
- Line 87: Please add the type of oil you used in this study
The type of oil ‘extra virgin olive oil’, has been added in the text (L89).
- Line 99: delete “and the two types of HPMC (Biosynth carbosynth Limited, UK)” it was written in line 92.
The last part of the sentence was deleted to avoid repetition (L101).
- Line 112: How the HPMC solution was prepared and when it was added to emulsions?
The method followed to prepare HMPC solutions is described in section ‘2.3. Preparation of HPMC solutions’ (L107-114):
‘Concentrations of HPMC solutions were selected after preliminary work was car-ried out to achieve HPMC solutions that exhibit viscoelastic moduli values within the same range. HPMC solutions (4, 6, 8 and 10% w/w of HPMC-L and 2, 3, 4 and 5% w/w of HPMC-H) were prepared following the method described by Liu, et al. [36] and Ding, et al. [37] with some modifications. HPMC powder was dispersed into distilled water using a high-speed homogeniser (Silverson, Model L4RT, UK) at 200 rpm for 30 min at room temperature, then left at 4 °C for at least 24 h in order to allow the HPMC to be completely hydrated before the measurements.’
To prepare the emulsions stabilised with HMPC and lecithin HPMC was added to the emulsion as describe in L132-142:
‘HPMC-L was then added at three different concentrations (0, 2 and 4% w/w) to conventional emulsions (CE-0, CE-2, CE-4, respectively. The mixtures were stirred using a magnetic stirrer (ChemLab, Model SS3H) at 200 rpm for 3 h at ambient temperature to ensure complete dispersion, then left at 4 °C for at least 24 h in order to allow the HPMC to be completely hydrated before the measurements (Liu et al., 2008; Ding et al., 2014).
HPMC-L was added at two different levels (0% and 2%) to NE (NE-0 and NE-2, respectively). The NE was processed through a high-pressure homogeniser (HPH) (8.30H, Rannie, APV, Denmark) at 400 bars for 1 cycle.’
- Lines 121-127: Paraphrase and delete the repeated sentences.
This section has been modified following reviewers’ suggestions (L125-135).
- Line 136: ... were left for …
The sentence has been corrected (L144).
- Line 241: are
The sentence has been corrected (L251).
- Line 281: G' and G" moduli of HPMC solutions and butter as a function of frequency
The sentence has been corrected in Figure 1 caption (L291).
- Table 2: The statistical analysis for “Work of shear” is not correct.
Values of work of shear in Table 2 were wrong and have now been rectify (L368). After this correction, the discussion of firmness and work of shear of the HMPC solutions in section 3.1.3. follows the trend presented in Table 2 (L359-367).
- Line 575: (0, 2 and 4 %)
The sentence has been corrected in Figure 8 caption (L589).